# Genome-Wide Characterization and Identification of Auxin Response Factor (*ARF*) Gene Family Reveals the Regulation of RrARF5 in AsA Metabolism in *Rosa roxburghii* Tratt. Fruits

**DOI:** 10.3390/biology14091156

**Published:** 2025-09-01

**Authors:** Tu Feng, Zhengliang Sun, Mingchun Liu, Hong Zhao, Yizhong Zhang, Pedro Garcia-Caparros, Bin Yang, Yingdie Yang

**Affiliations:** 1School of Ecological Engineering, Guizhou University of Engineering Science, Bijie 551700, China; fengtu@gues.edu.cn (T.F.); zhaohonggzu@163.com (H.Z.); z8300300@126.com (Y.Z.); 2Key Laboratory of Bio-Resource and Eco-Environment of Ministry of Education, College of Life Sciences, Sichuan University, Chengdu 610065, China; sunzhengliang99@163.com (Z.S.); mcliu@scu.edu.cn (M.L.); 3Higher Engineering School, University of Almeria, 04120 Almeria, Spain; pedrogar123@hotmail.com; 4Deyang Guoke Dual Carbon Research Institute, Deyang 618221, China; yangbin@inno-bio.org

**Keywords:** *R. roxburghii*, ARF, genome-wide identification, fruit development, Vitamin C

## Abstract

*Rosa roxburghii* is recognized for its exceptional nutritional profile, particularly its high content of Vitamin C. Understanding the regulation mechanism of Vitamin C biosynthesis is vital for nutrition quality improvement in *R. roxburghii*. Auxin response factor (*ARF*) family members are key transcription factors involved in various fruit development processes and nutrition quality formation. However, ARF family genes have not yet been identified at the genome-wide level in *R. roxburghii*. In the present study, we identified 14 auxin response factor (*ARF*) family genes, which can be divided into four subfamilies. We found that one ARF family member, RrARF5, acts as a putative regulator in the modulation of Vitamin C accumulation through the activation of the expression of *RrMDHAR1*, a key enzyme involved in Vitamin C biosynthesis. Our findings extend our understanding of Vitamin C biosynthesis regulation and provide a putative target for fruit nutrition improvement.

## 1. Introduction

Auxin is a significant plant hormone that plays a pivotal role in various plant developmental processes, including embryogenesis, vascular bundle formation, flower and fruit development, and root growth [1]. The transcriptional regulation of several gene families, such as Gretchen Hagen 3 (GH3), Aux/IAA, and auxin response factor (ARF) families, has been shown to influence auxin synthesis and transport, thereby affecting overall plant growth and development [2,3,4]. Of these families, ARF plays a significant role in regulating plant growth and development by binding to the TGTCTC DNA motif, mediating hormonal responses, and consequently affecting plant developmental processes [5].

A typical ARF consists of three distinct structural domains. The first is the highly conserved N-terminal B3-like DNA-binding domain (DBD), which is present in all ARFs and is essential for binding to target DNA sequences. The second domain is the C-terminal dimerization domain (CTD), which facilitates the formation of heterodimers with proteins from the Aux/IAA family. The third domain is the variable intermediate region (MR), which can act either as an activation domain (AD) or a repression domain (RD) [6,7,8]. The regulatory activity of ARF depends on the proper presence and interaction of all three domains [9].

*ARF* gene families have been identified in numerous plant species, including *Arabidopsis thaliana* L. [10], *Fragaria vesca* L. [11], *Vitis vinifera* L. [12], *Citrus sinensis* L. [13], *Capsicum annuum* L. [14], *Solanum lycopersicum* L. [3], and *Oryza sativa* L. [9]. *ARF* family genes play critical roles in various developmental and physiological processes in plants, such as embryogenesis [15], lateral root growth [16], leaf expansion [17], and leaf senescence [18]. In *Arabidopsis thaliana*, the auxin response factor AtARF3/4 is functionally implicated in floral organ morphogenesis, thereby modulating floral development and organogenetic processes [19]. In addition, *ARF* gene family members have also been shown to function as key regulators in fruit development, ripening, and quality metabolism [20,21,22,23,24,25]. MdARF5 was reported to activate the expression of ethylene-related genes, thereby initiating apple fruit ripening [26]. Similarly, CpARF2 is involved in the regulation of fruit ripening in papaya by promoting the transcriptional activity of *CpEIL1* [27]. In tomato, SlARF10 and SlARF6A have been demonstrated to facilitate chlorophyll accumulation by activating *SlGLK1* expression [24,25]. However, the role and mode of action of ARF in Vitamin C (Vc) biosynthesis and metabolism remain largely unknown.

The wild fruit crop *R. roxburghii*, belonging to the genus *Rosa* within the family Rosaceae, is widely distributed in the mountainous regions of southwestern and south-central China. Its cultivation has expanded significantly, with an estimated planting area exceeding 30,000 hectares and an annual yield of approximately 2000–3000 tons [28]. The fruit is edible and characterized by a unique aroma, crunchy texture, and a slightly acidic and astringent flavor [29]. *R. roxburghii* is recognized for its exceptional nutritional profile, particularly its high content of ascorbic acid (Vitamin C) [30]. Ascorbic acid is a vital metabolite for most organisms, serving as an antioxidant and cofactor in various biological processes, including stress resistance, phytohormone biosynthesis, cell division, and cell expansion [31]. During plant growth, AsA accumulation is primarily regulated by Vc biosynthesis, metabolism, and translocation. So far, four major biosynthetic pathways have been identified in plants: the L-galactose pathway, the inositol pathway, the D-galacturonic acid pathway, and the L-glucose pathway. AsA metabolism is further divided into degradation and recycling pathways. Within the L-galactose pathway, L-galactose dehydrogenase (GalDH) functions as a key precursor enzyme, while L-galactono-1,4-lactone dehydrogenase (GalLDH) catalyzes the final step, converting L-galactono-1,4-lactone into AsA. Two additional enzymes, GDP-L-galactose phosphorylase (GGP) and GDP-D-mannose 3′,5′-differential isomerase (GME), are critical regulatory components of this biosynthetic pathway. AsA also plays an essential physiological role in scavenging reactive oxygen species (ROS), serving as a specific electron donor in the reduction of hydrogen peroxide (H_2_O_2_) to H_2_O and O_2_. The recycling of oxidized AsA is mediated by monodehydroascorbate reductase (MDHAR) and dehydroascorbate reductase (DHAR), which are crucial for maintaining cellular AsA levels. In previous studies on kiwifruit, the enzymatic activities of GalDH, GalLDH, MDHAR, and DHAR showed a similar trend of AsA accumulation, with significantly positive correlations observed between enzyme activity and AsA content, suggesting coordinated regulation [32]. Since humans and many other animals lack the endogenous capacity to synthesize AsA, it must be obtained from dietary sources like fresh fruits and vegetables [33]. The pattern of AsA accumulation varies among fruit species: in kiwifruit and apple, peak AsA levels are observed during the young fruit stage; in jujube, during the fruit expansion stage; and in strawberry and tomato, during the fruit ripening stage [34]. Previous studies have shown that AsA content increases progressively in *R. roxburghii* throughout fruit development and the ripening process [35]. Due to these characteristics, *R. roxburghii* is renowned for its exceptionally high AsA concentration, earning it the title “King of Vitamin C” and becoming a commercially significant fruit crop. Therefore, elucidating the regulatory mechanisms underlying AsA accumulation during *R. roxburghii* fruit development holds substantial significance for both crop improvement and nutritional enhancement. Despite *ARF* genes being well studied in several fruit crops, their genomic organization, expression profiles, and specific regulatory roles in Vitamin C metabolism in *Rosa roxburghii* remain uncharacterized.

Recently, genome-wide data for *R. roxburghii* have been assembled [35]. However, despite the fact that the ARF gene family is widely studied in other species, its role in *R. roxburghii* remains unexplored. Given the ARF family’s known roles in fruit development and ripening, it is likely that these genes play a similar role in *R. roxburghii*. Given the significant implications of ascorbic acid (ASA) for human health, elucidating the regulatory mechanisms underlying Vitamin C biosynthesis and metabolism in *R. roxburghii* fruits has emerged as a research priority. In this study, we performed the first comprehensive genome-wide analysis of the *auxin response factor* (*ARF*) gene family in *R. roxburghii*. A total of 14 *RrARF* genes were identified and classified into four distinct subgroups. These genes were comprehensively characterized, including their chromosomal localization, phylogenetic relationships, gene structures, and cis-regulatory elements. Furthermore, RNA-Seq data were systematically analyzed to delineate the spatiotemporal expression patterns of *RrARF* genes across various tissues and developmental stages. Through an integrated approach encompassing coexpression network analysis, transcriptional activation assays, and transient overexpression experiments, we demonstrate that RrARF5, a member of the *R. roxburghii* ARF family, functions as a positive regulator of Vitamin C (Vc) biosynthesis by transcriptionally activating the structural gene *RrMDHAR1*. These findings not only advance our understanding of the functional roles of RrARF5 but also provide critical insights into the molecular mechanisms governing Vc accumulation in plant fruits via the *ARF* gene family. This work establishes a foundational framework for future research aimed at enhancing fruit quality in *R. roxburghii* cultivars through targeted genetic improvement.

## 2. Materials and Methods

### 2.1. Identification of ARF Family Genes in R. roxburghii Genome

The *R. roxburghii* genome sequence was obtained from the CNSA (CNGB Nucleotide Sequence Archive) with accession number CNP0004212 (https://db.cngb.org/cnsa/, accessed on 28 October 2024) [35]. To effectively identify members of the ARF family in the *R. roxburghii* genome, candidate protein sequences containing the auxin response (PF06507) and B3 DNA-binding domain (PF02362) structural motifs were downloaded from the Pfam protein family database (http://Pfam.xfam.org/, accessed on 29 October 2024). The ‘Simple HMM search’ function in TBtools (Version 2.210) software [36] was used to perform a BLAST search on the ARF protein sequences of *R. roxburghii* using PF06507 (*p* < 0.001) and PF02362 (*p* < 0.001) as query models (*e*-value ≤ 1 × 10^−10^). Candidate *ARF* genes were identified from the *R. roxburghii* genome. The NCBI-CDD (https://www.ncbi.nlm.nih.gov/cdd, accessed on 2 November 2024) and SMART (http://smart.embl.de/, accessed on 2 November 2024) databases were used to confirm each putative RrARF; specifically, through the SMART database (http://SMART.embl-heidelberg.de/, accessed on 2 November 2024), Markov model (HMM) files were utilized. The number of amino acid residues (AA), molecular weight (MW), and theoretical the isoelectric point (pI) of the RrARF proteins were calculated using ExPASy (Version 2.210) software (https://www.expasy.org/, accessed on 2 November 2024).

### 2.2. Chromosomal Distribution of RrARFs

The locations of *RrARFs* on chromosomes were extracted from the genome annotation files. The gene densities across the entire chromosomes were determined and visualized using TBtools software [36]. For further analysis of conserved motifs and domains, only *RrARFs* that were anchored to chromosomes were displayed.

### 2.3. Analysis of Cis-Acting Elements in RrARF Promoters

The 2000 bp sequences upstream of the initiation codon (ATG) of each *R. roxburghii ARF* gene (RrARF) were extracted from the *R. roxburghii* genome database using TBtools software [36]. The PlantCARE database was used to further analyze the cis-acting regulatory elements presented in the promoter regions of *ARF* family genes (http://bioinformatics.psb.ugent.be/webtools/plantcare/html/, accessed on 4 November 2024).

### 2.4. Phylogenetic Analysis of the R. roxburghii ARF Gene Family

The ARF protein sequences from *R. roxburghii* (14 sequences), *Arabidopsis thaliana* (23 sequences), *Fragaria vesca* (12 sequences), *Malus domestica* (29 sequences), and *Vitis vinifera* (19 sequences) were subjected to multiple sequence alignments using the default settings of Clustal W in MEGA 7 (Version 2.210) software. A phylogenetic tree was then constructed based on the results of the sequence alignment using maximum likelihood [37]. The protein sequences of ARFs from *Arabidopsis thaliana*, *Fragaria vesca*, *Malus domestica*, and *Vitis vinifera* species were obtained from the NCBI database (https://www.ncbi.nlm.nih.gov/, accessed on 4 November 2024). The final phylogenetic tree was edited and visualized using the iTOL online tool (https://itol.embl.de/, accessed on 4 November 2024).

### 2.5. Gene Structure and Conserved Motif Characterization

Conserved motifs in RrARFs were identified using the MEME server (https://meme-suite.org/, accessed on 28 November 2024) with the following parameters: a maximum number of 10 motifs, a minimum motif width of 6, and a maximum motif width of 50. We performed structural domain analysis using the NCBI Conserved Domain Database (CDD) to ascertain the types and location of structural domains within all RrARF sequences. In addition, we used TBtools software to visualize the exon/intron structures of the *RrARF* genes, as well as the conserved motifs and structural domains of the corresponding RrARF proteins [36].

### 2.6. Gene Expression Analysis of ARF Family Genes

RNA-Seq data from different tissues and organs of *R. roxburghii*, including root, stem, leaf, flower, and fruits harvested in different developmental and ripening stages, were downloaded from the CNSA (https://db.cngb.org/cnsa/, accessed on 28 November 2024) under accession number CNP0004212 [35]. Data filtering and quality control were performed using fastp (Version 1.0.1) software, and the resulting clean data were used for subsequent analyses [38]. The *R. roxburghii* genome was used as a reference for read alignment (https://db.cngb.org/cnsa/, accessed on 28 November 2024), and String Tie was used to quantify the aligned reads. The differential expression TPM (transcripts per kilobase of exon model per million mapped reads) across different tissues was calculated using TBtools [36]. The normalized data were then used to plot expression heat maps with TBtools.

### 2.7. Weighted Correlation Network Analysis (WGCNA)

The top 50% of genes, ranked according to their expression levels, were selected for WGCNA. After an initial threshold screening, a soft-threshold power of β = 7 was selected to strengthen the original proportionality matrix to obtain the derivation of an adjusted neighbor-joining matrix. To better assess the correlation of expression patterns among genes, the neighbor-joining matrices were further transformed into topological overlap matrices (TOMs). Subsequently, gene-shearing analysis was performed using a topological heterogeneity dissimilarity matrix, defined as dissTOM = 1 − TOM. Clustering and module partitioning were then performed, with a minimum module size set at thirty genes (minModuleSize = 30). Module merging was based on a similarity threshold of 0.3 (cutHeight = 0.3), and the network type was defined as “signed” (type = “signed” or networkType = “signed”). The network was visualized using Cytoscape (Version 3.8.0) software [39].

### 2.8. Dual-Luciferase (Dual-LUC) Reporter Assay

To investigate the transcriptional regulation of *RrMDHAR1* by RrARF5, the coding sequence of RrARF5 was cloned into the pGreenII 62SK vector to function as the effector construct. The promoter sequence of *RrMDHAR1* was cloned into the pGL3 vector as the reporter construct. Leaf protoplasts from *N. benthamiana* were then prepared following the methodology reported by Pei [40]. The dual luciferase reporter assay kit was used to quantify the transcriptional activity, according to the manufacturer’s instructions.

### 2.9. Accession Numbers

The accession numbers of *VvARF* genes in *Vitis vinifera* were reported in the study of Wan [12]. The accession numbers of *AtARF* genes in *Arabidopsis thaliana* and *MdARF* genes in *Malus domestica* were reported in the study of Luo [41]. The accession numbers of *FvARF* genes in *Fragaria vesca* were reported in the study of Zhang [11].

## 3. Results

### 3.1. Identification of ARF Family Genes in R. roxburghii Genome

Following an HMMER search and subsequent analysis using the NCBI Conserved Domain Database (CDD) search tool, 14 *ARF* genes were identified in the *R. roxburghii* genome (Table 1). Each of the corresponding RrARF proteins contained the characteristic structural domains of ARF proteins (auxin response factor, Auxin Resp). These genes were then sequentially named *RrARF1* to *RrARF14* based on their chromosomal locations (Figure 1, Table 1).

The amino acid (aa) lengths of the RrARF proteins vary from 570 aa (RrARF11) to 1182 aa (RrARF1), resulting in an average protein length of 791.64 aa. Accordingly, the molecular weights of the RrARF proteins ranged from 62.6 kDa (RrARF11) to 132.8 kDa (RrARF1). The isoelectric points for the 14 RrARF proteins were mainly below 7, except for RrARF6, which exhibited a higher pl of 8.56, suggesting that RrARF proteins may function in diverse cellular microenvironments.

To investigate the evolutionary relationships among *RrARF* homologous genes, 97 ARF proteins were obtained from five different plant species: *R. roxburghii*, *Arabidopsis thaliana*, *Malus domestica* (apple), *Vitis vinifera* (grape), and *Fragaria vesca* (wild strawberry). A phylogenetic tree was constructed using the neighbor-joining method, which is presented in Figure 2. Based on the characteristics of the protein structural domains of the ARF family in *Arabidopsis thaliana*, our phylogenetic tree classified these ARF homologues into four distinct groups. The distribution of the 14 RrARF members across these subgroups was as follows: subgroup I (1 member), subgroup II (4 members), subgroup III (5 members), and subgroup IV (4 members).

### 3.2. Analysis of Conserved Structural Domains and Promoter Sequences of ARF Family Members in R. roxburghii

To gain further insight into the structural characteristics of the RrARF members and to ascertain their potential function, an in-depth examination was conducted on their exon-intron configuration, conserved structural domain, and motif composition (Figure 3).

To fully confirm the accuracy of the conserved structural domains in RrARF, a multiple-sequence comparison was performed using ClustalW. The typical ARF structure is characterized by the presence of three conserved structural domains: B3 (DBD), Auxin_responsive, and AUX/IAA superfamily (CTD) [42]. The results showed that all members of the RrARF family possessed both the B3 structural domain (DBD) and the auxin-responsive structural domain. However, the AUX/IAA domain (CTD) was lacking in *RrARF4*, *RrARF6*, *RrARF11*, *RrARF12*, and *RrARF14* (Figure 3C and Appendix A).

In addition, the RrARF protein sequence was further analyzed using the online software MEME, and 12 motifs ranging from 21 to 50 amino acids in length were identified (Figure 3B; Appendix A). Among these motifs, motifs 1, 2, 3, and 5 belong to the B3 structural domain (DBD); motifs 6, 8, and 11 belong to the auxin response structural domain; and motifs 9 and 10 belong to the AUX/IAA structural domain (CTD).

Gene structure analysis revealed that the majority of the coding sequences were interrupted by introns, with members of the same phylogenetic group often sharing similar gene structures (Figure 3D). Interestingly, although RrARF9 and RrARF10 from the class III group are similar in their protein structures, their exon–intron distributions showed differences. In addition, although RrARF1 and RrARF5 from the class IV group have the same number of exons–introns, their CDS sizes are different. These differences may indicate that RrARFs have diverse splicing and rich transcripts.

Given the important role of *cis*-acting elements in regulating gene expression, we retrieved the 2 kb promoter sequences of all the identified *RrARF* genes and submitted them to the PlantCare database for cis-acting element prediction and analysis. This analysis revealed the presence of hormone-responsive elements (e.g., MeJA, abscisic acid, salicylic acid, gibberellin, etc.), wound-responsive elements, and stress-related elements (e.g., low temperature, drought, etc.), as well as light-responsive elements (Appendix A). The TGA-element and AuxRR-core motifs, which are indicative of a typical auxin response, were only identified in the promoters of *RrARF1*, *RrARF7*, *RrARF9*, and *RrARF13*, suggesting a putative role for these genes in auxin-mediated growth processes.

### 3.3. Expression Profiles of RrARFs in Different Tissues and Developmental Stages

To investigate the potential functions of the *RrARF* gene family in *R. roxburghii*, we analyzed the spatiotemporal expression profiles using published RNA-seq data from various tissues and different developmental stages of *R. roxburghii*, including root, stem, leaf, and flower, as well as fruit tissues sampled at five different developmental and ripening stages. The expression profiles of the 14 *RrARF* family genes in different tissues and during fruit developmental stages can be divided into four subgroups (Figure 4). The four genes of *RrARF* that make up subgroup I (*RrARF5*, *RrARF11*, *RrARF12*, and *RrARF14*) were highly expressed in the latter stages of fruit growth and ripening, indicating a potential role in the processes of late fruit maturity and ripening. Two *RrARF* genes (*RrARF1* and *RrARF6*) that are mostly expressed in leaf and flower tissues make up subgroup II. This suggests that the two genes may have a role in leaf and flower development. Four genes of *RrARF* (*RrARF7*, *RrARF8*, *RrARF9*, and *RrARF10*) found in subgroup III displayed strong expression in roots, stems, and young fruits, suggesting that these members may be essential to the development of both vegetative and early fruit. Subgroup IV contains four *RrARF* genes (*RrARF2*, *RrARF3*, *RrARF4*, and *RrARF13*) with specific expression patterns in the root and stem, indicating their vital function in root and stem growth and development.

### 3.4. Coexpressed Gene Networks of RrARFs

To investigate the expression patterns of *RrARF* genes associated with Vitamin C synthesis and metabolism, a coexpression network was constructed using weighted gene coexpression network analysis (WGCNA). The expression profiles of 14,310 genes were grouped into 14 modules, with the remaining unclustered genes placed in gray modules (Figure 5A and Appendix A). The number of genes within each module is provided in Appendix A. The turquoise module was the most gene-rich, comprising 3890 genes. In contrast, the module with the lowest number of genes was the cyan module, which contained 49 genes (Appendix A).

The correlation between the 14 modules and different periods of fruit development was analyzed (Figure 5B). Among these, the MEblue module showed the highest correlation with the fruit-oct stage (r = 0.95, *p* < 0.000000081), which is a key phase for Vitamin C synthesis in *R. roxburghii*. Consequently, the *RrARF* genes within the blue module were subsequently identified as the core genes in Vc biosynthesis regulation. A gene coexpression network was then constructed using these core genes and their coexpression genes related to Vitamin C synthesis and metabolism (Figure 5C). As shown in Figure 5C, three ripening-related *ARF* genes (*RrARF5*, *RrARF11*, and *RrARF12*) showed a high correlation with five genes involved in VC biosynthesis, suggesting a putative regulatory relationship between these *ARFs* and the five VC biosynthesis genes. The weight values between two genes in the gene interaction network are shown in Appendix A.

### 3.5. RrARF5 Involved in VC Biosynthesis by Activating RrMDHAR1 Transcription

The analysis of the public RNA-seq data enabled the examination of the expression of genes associated with Vitamin C synthesis and metabolism within the context of a coexpression network. The results demonstrated that gene *Rr602006,* named *RrMDHAR1*, which encodes monodehydroascorbate reductase (MDHAR) (Appendix A), exhibited high expression levels in *R. roxburghii* during fruit development (Appendix A). Additionally, the expression patterns of *RrARF5* and *RrMDHAR1* exhibited a high positive correlation during fruit development and ripening (Figure 5C and Appendix A), suggesting a positive regulatory relationship between *RrARF5* and *RrMDHAR1*. Promoter analysis revealed the presence of two TGA elements and an AuxRR core ARF transcription factor regulatory element in the promoter region of *RrMDHAR1* (Appendix A). Therefore, it was hypothesized that *RrMDHAR1* may be a target of *RrARF5*. To further investigate their relationship, we performed a dual-luciferase reporter assay in *Nicotiana benthamiana* leaf protoplasts by co-transfecting effector construct *35S:RrARF5* with reporter construct *pro:RrMDHAR1* (Figure 6A). The results showed that RrARF5 significantly activated the expression of *RrMDHAR1* compared to the control (Figure 6B), thereby supporting the hypothesis that RrARF5 is critically involved in Vc biosynthesis by regulating the expression of *RrMDHAR1* in *R. roxburghii* fruits.

To further validate the regulation of RrARF5 in Vc accumulation by regulating *RrMDHAR1*, we constructed *35S:RrMDHAR1-FLAG* and *35S:RrARF5-FLAG* vectors and transiently expressed these vectors in tomato fruits since the genetic transformation and transient expression are currently not feasible for *R. roxburghii* (Figure 6C). As shown in Figure 6D, the ectopic expression of *35S:RrMDHAR1-FLAG* in tomato fruit resulted in a significant increase in Vc content compared to the control (injected with an empty vector). Moreover, the ectopic expression of both *35S:RrMDHAR1-FLAG* and *35S:RrARF5-FLAG* in tomato fruits led to significantly higher content of Vc compared to the fruits with only an ectopic expression of *35S:RrMDHAR1-FLAG* (Figure 6D). These data further indicate that RrARF5 modulates Vc accumulation in fruits by regulating the expression of *RrMDHAR1* in *R. roxburghii*.

## 4. Discussion

The ARF family is a central component of the auxin signaling pathway, regulating a multitude of processes integral to plant development. The ARF transcription factor family has been extensively studied in various plant species, such as *Arabidopsis thaliana* [6], tomato (*Solanum lycopersicum*) [43], papaya (*Carica papaya* L.) [44], longan (*Dimocarpus longan* L.) [45], apple (*Malus domestica*) [41], and rice (*Oryza sativa*) [9]. Nevertheless, comprehensive investigations of the entire *ARF* gene family in *R. roxburghii* have not yet been conducted. In the present study, we identified 14 RrARF transcription factors in *R. roxburghii*, which were unevenly distributed across seven chromosomes (Figure 1). Chromosome 5 harbored the highest number of RrARF transcription factor members (four), whereas chromosomes 1, 4, and 6 each contained a single RrARF transcription factor member. Phylogenetic analysis of the 14 RrARF proteins showed that they can be classified into four major groups, with each group showing a close evolutionary relationship to the *FvARF* genes (Figure 2). This observation indicates that *ARF* genes in these taxa may have originated from a common ancestor, or alternatively, that they may share highly conserved functions [46]. Moreover, we identified twelve conserved motifs in the RrARF family in *R. roxburghii* (Figure 3). Although the number of members differs in each phylogenetic group, the motifs within a group are markedly conserved. In addition to the conserved motifs, further research is required to determine whether additional motifs may be associated with novel functions.

The majority of ARFs comprise three structural domains with conserved features [43]. However, five RrARFs (RrARF4, RrARF6, RrARF11, RrARF12, and RrARF14) lack the Aux/IAA-binding domain (Appendix A), which is consistent with the structure of the AtARF3 from the Arabidopsis subfamily that lacks the CTD structural domain. Moreover, the ARF members that lack the Aux/IAA-binding domain can also be found in tomato, rice, and orchid species [47,48,49]. Since the Aux/IAA-binding domain is critical for the interaction between ARF and Aux/IAA, the lack of this domain in the five RrARFs may suggest these proteins exert their function in an auxin-dependent manner. It is noteworthy that RrARF7 contains a PABP-1234 superfamily structural domain in addition to the three typical ARF domains (Figure 3). However, the function of this conserved domain and that of RrARF7 require further study. Additionally, the presence of cis-acting elements associated with plant development, abiotic and biotic stresses, light responses, and phytohormone responses in the promoter regions of the *RrARF* family genes and their distinct expression patterns in different tissues and fruit developmental stages support their diverse functions in plant growth and fruit development. However, the functional significances of these *RrARF* genes require further investigation.

High Vitamin C content is one of the most important quality traits of fruits of *R. roxburghii*, and understanding the regulatory mechanism of Vitamin C biosynthesis and metabolism is vital for improving the fruit quality of *R. roxburghii*. In the present study, through coexpression analysis, transactivation assays, and transient overexpression assays, we revealed that RrARF5 acts as a positive regulator of Vc biosynthesis by activating the expression of *RrMDHAR1,* a structural gene reported to be correlated with ascorbic acid content [50]. Further investigation is required to elucidate the molecular mechanism underlying the function of RrARF5. Phylogenetic analysis reveals that RrARF5 exhibits a higher degree of sequence homology with FvARF6, a member of the *auxin response factor* (*ARF*) gene family in strawberry (Fragaria × ananassa) belonging to the Rosaceae family (Figure 4). Previous studies have demonstrated that the regulatory function of ARF proteins—whether acting as transcriptional activators or repressors—can be inferred based on the amino acid composition of their middle region (MR) domains [11]. Using this criterion, FvARF6 was predicted to function as a potential transcriptional repressor. In the present study, we experimentally confirmed that RrARF5 acts as a transcriptional activator through a dual-luciferase reporter assay (Figure 6B), highlighting the need for more comprehensive research to determine the specific regulatory mechanisms. Similarly to FvARF6, RrARF5 demonstrates elevated expression levels during the early stages of fruit development, suggesting potential functional similarities in floral and other reproductive organs (Figure 4). These findings contribute to a better understanding of the function of the *ARF* gene family in fruit quality regulation, while also providing a theoretical foundation for further research into the molecular mechanisms underlying ascorbic acid accumulation in *R. roxburghii*.

It is worth pointing out that although we identified the *ARF* gene family in *R. roxburghii* and revealed the role of one ARF member in Vc accumulation, the functions of other *ARF* genes in different plant growth and developmental processes remain largely unknown. Future research should focus on developing genetic transformation systems for *R. roxburghii* to experimentally validate the roles of ARF family members.

## 5. Conclusions

Through genome-wide identification and analysis of auxin response factor (*ARF*) family genes in *R. roxburghi*, we identified 14 *ARF* family genes (designated as *RrARFs*), which are grouped into four subfamilies. These genes exhibit differential expression patterns across different tissues and stages during the fruit development and ripening processes, with four members displaying ripening-related expression. *RrARF5* was found to be highly coexpressed with *RrMDHAR1*, a key enzyme involved in Vitamin C biosynthesis. Transactivation assays and transient overexpression experiments demonstrated the involvement of RrARF5 in Vitamin C accumulation by activating the transcription of *RrMDHAR1*. The outcome of this study suggests a potential role of the *ARF* gene family in Vitamin C biosynthesis in *R. roxburghii* fruits and provides a putative target for Vitamin C improvement.

## Figures and Tables

**Figure 1 biology-14-01156-f001:**
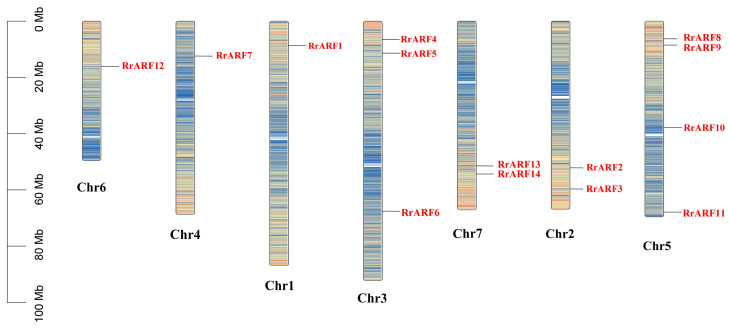
**Distribution of ARF family genes on *Rosa roxburghii* chromosomes.** The gene density was visualized with the color intensity.

**Figure 2 biology-14-01156-f002:**
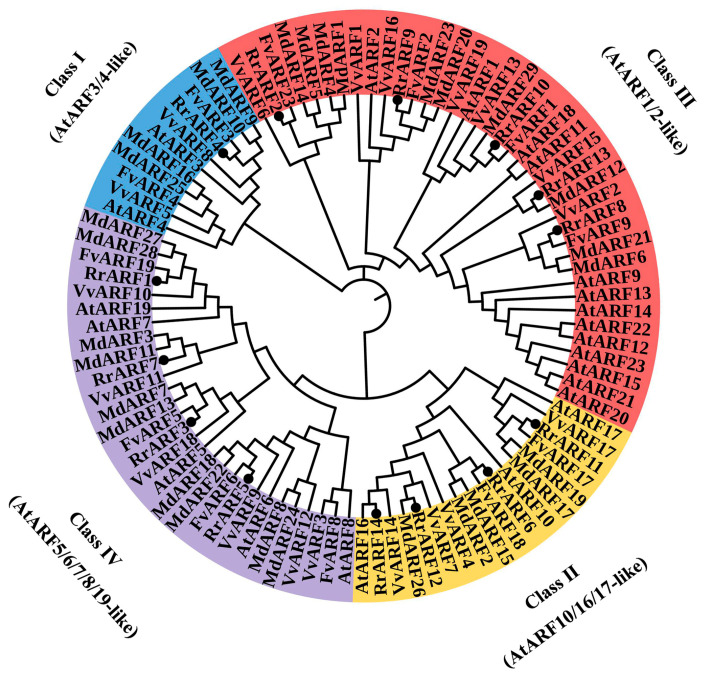
**Phylogenetic analysis of *ARF* family genes in different plant species.** The *ARF* family genes are divided into four groups, represented by different colors. At, *Arabidopsis thaliana*; Fv, *Fragaria vesca*; Md, *Malus domestica*; Rr, *Rosa roxburghii*; and Vv, *Vitis vinifera*.

**Figure 3 biology-14-01156-f003:**
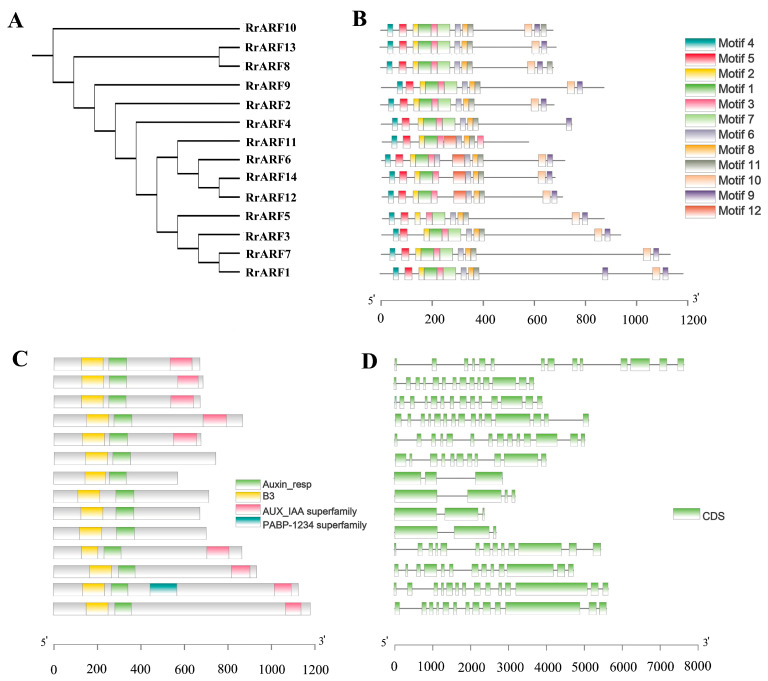
**Conserved motifs, typical ARF structures, and gene structure analysis of *Rosa roxburghii ARF* family genes:** (**A**) Phylogenetic tree of 14 *ARF* genes from *Rosa roxburghii*. (**B**) Distribution of conserved motifs in RrARF proteins. (**C**) Analysis of RrARF protein domains. (**D**) Exon–intron structure of the *RrARF* members. Different colored boxes represent different themes. The scale bars are shown at the bottom.

**Figure 4 biology-14-01156-f004:**
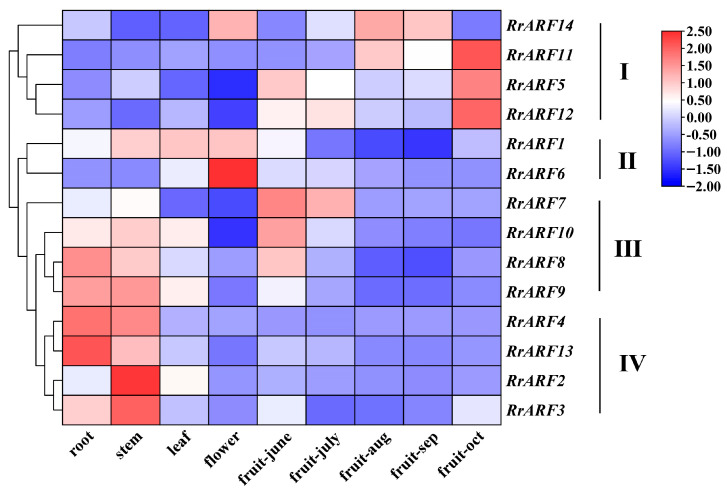
**Expression patterns of *RrARF* genes in different tissues.** The heatmap shows the expression patterns of *RrARFs* in various tissues and fruit developmental stages, as derived from transcriptomic data.

**Figure 5 biology-14-01156-f005:**
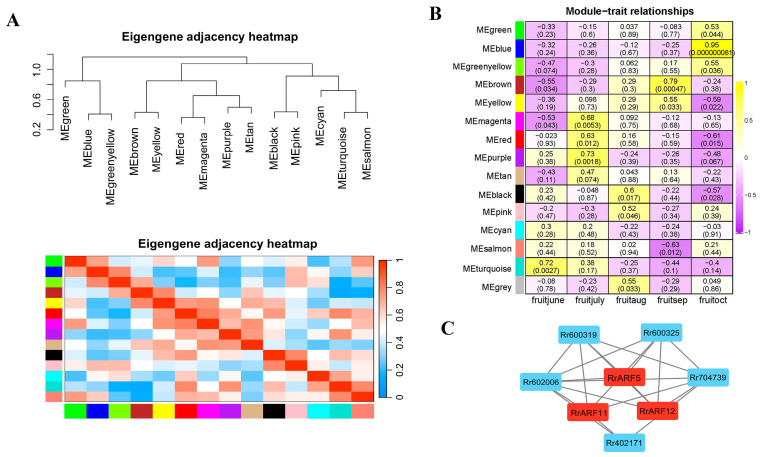
**WGCNA of *RrARF* genes and genes related to Vitamin C synthesis and metabolism in *R. roxburghii*:** (**A**) Gene clustering dendrogram in WGCNA. The upper part shows the gene clustering dendrogram, and the lower part shows the assigned modules with the same colors for the same modules. (**B**) The relationships between the identified modules and samples at different stages of fruit development were visualized using heatmaps. The numbers displayed within the boxes represent the correlation coefficients and corresponding *p*-values between the identified modules and the relevant samples. (**C**) The coexpression network of *RrARF* genes and genes related to Vitamin C synthesis and metabolism in *R. roxburghii* within the blue module.

**Figure 6 biology-14-01156-f006:**
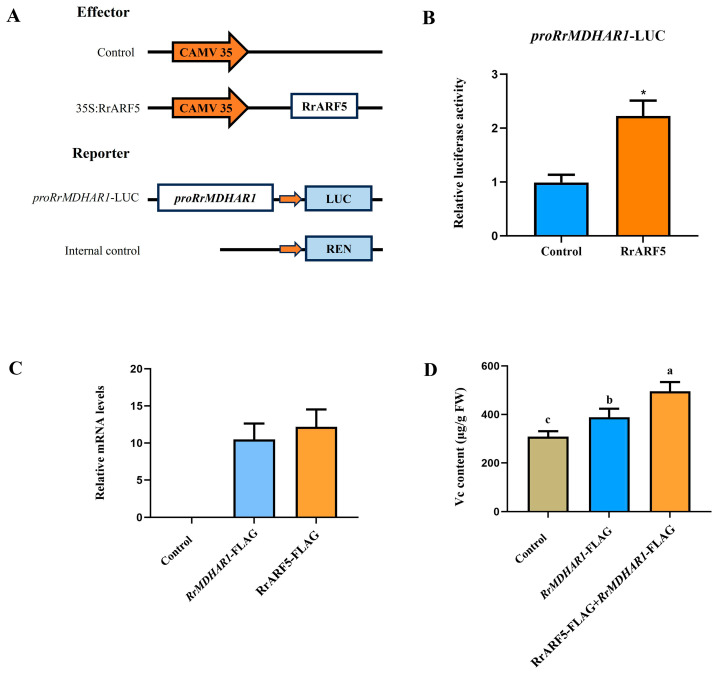
**RrARF5 regulates Vc accumulation by activating the expression of the *RrMDHAR1* gene:** (**A**) Schematic of effector and reporter constructs used in transient expression assays. (**B**) The activation effect of RrARF5 on the transcription of *RrMDHAR1*. Asterisks indicate statistical significance using Student’s *t* test, *p* < 0.05. (**C**) Transcript accumulation of chimeric *RrMDHAR1-FLAG* and *RrARF5-FLAG* in the transient expression in fruits of tomato. Transcript levels were assessed via qRT-PCR in the pericarp of injected tomato fruits. The Sl-Actin gene was used as an internal control. (**D**) The content of Vitamin C in the fruits of control, injected with *35S:RrMDHAR1-FLAG* construct, and injected with both *35S:RrMDHAR1-FLAG and 35S:RrARF5-FLAG* tomato. Control, injected with an empty vector. FW, fresh weight. Values are means ± SD of three replicates. Different lowercase letters indicate significant differences.

**Table 1 biology-14-01156-t001:** Characterization of *RrARF* genes in *Rosa roxburghii*.

Gene	Gene	CDS	Chromosome	Protein		
Name	ID	Size	Location	Length (aa)	MW (KDa)	pI
*RrARF1*	Rr100998	3549	Chr1	1182	132.8	6.15
*RrARF2*	Rr203848	2031	Chr2	676	75.5	5.83
*RrARF3*	Rr204729	2799	Chr2	932	102.6	5.25
*RrARF4*	Rr300761	2232	Chr3	743	81.5	6.31
*RrARF5*	Rr301243	2598	Chr3	865	95.3	6.09
*RrARF6*	Rr304962	2145	Chr3	714	78.7	8.56
*RrARF7*	Rr400968	3384	Chr4	1127	124.7	6.24
*RrARF8*	Rr500747	2019	Chr5	672	74.6	6.23
*RrARF9*	Rr500973	2607	Chr5	868	96.5	6.07
*RrARF10*	Rr503508	2016	Chr5	671	74.4	6.10
*RrARF11*	Rr505360	1713	Chr5	570	62.6	6.13
*RrARF12*	Rr601745	2112	Chr6	703	77.3	6.57
*RrARF13*	Rr703521	2061	Chr7	686	76.0	6.56
*RrARF14*	Rr703808	2025	Chr7	674	74.4	6.54

## Data Availability

The original contributions presented in this study are included in the article/Appendix A. Further inquiries can be directed to the corresponding author(s).

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
