# Peer review of "Genome-Wide Characterization and Identification of Auxin Response Factor (ARF) Gene Family Reveals the Regulation of RrARF5 in AsA Metabolism in Rosa roxburghii Tratt. Fruits"

_biology, 2025, doi:10.3390/biology14091156_

Round 1

Reviewer 1 Report

Comments and Suggestions for Authors

General comments (reviewer style)
The authors present a comprehensive characterization of the Auxin Responsive Factor (ARF) gene family in Rosa roxburghii Tratt., a species valued for its high vitamin C content but previously unstudied in this context. They identified 14 RrARF genes across seven chromosomes, classified into four subfamilies, and provided insightful analyses of cis-regulatory elements and gene expression patterns during fruit development. Importantly, the authors demonstrated a functional link between RrARF5 and RrMDHAR1, a key enzyme in vitamin C biosynthesis, through coexpression analysis and experimental validation via transactivation and overexpression assays. This study offers novel insights into the molecular mechanisms underlying fruit development and vitamin C accumulation in R. roxburghii, contributing valuable knowledge to the field.Try to avoid abbreviations in the manuscript title

The comments are as follows:

  • The Abstract should provide more information about the ARF gene family and its significance in flower development for this plant.
  • Be cautious with the phrase: “The outcomes of this study enhance our understanding of the ARF gene family's role in vitamin C accumulation in R. roxburghii and its broader impacts on fruit quality regulation.” This statement is too strong. Functional validation using transgenic plants is needed to firmly establish this connection. Your results suggest this relationship but do not conclusively prove it.
  • Avoid using words like “crucial or notably” as they are not typically scientific and may reflect AI translation artifacts.
  • Figures 1 and 3 require improvement and better integration into the text. Specifically, Figure 3 is difficult to interpret. Consider splitting it into parts A and B, and C and D below, enlarging the figure and providing a more detailed explanation in the manuscript.
  • Section 3.5 needs further development. The rationale should be clearer, and the results more thoroughly discussed. Also, Figure 5B should be enlarged, as it is currently hard to read.
  • The Discussion section can be strengthened by deeper analysis, particularly regarding gene function, chromosomal distribution, regulatory mechanisms, and transient expression data. It is important to explore all presented results comprehensively.

Comments on the Quality of English Language

can be improved, some terms are not scientific, and it may be from translator tools or AI

Reviewer 2 Report

Comments and Suggestions for Authors

Manuscript entitled: Genome-Wide Characterization and Identification of ARF Gene Family Reveals the Regulation of RrARF5 in AsA Metabolism in Rosa roxburghii Tratt. Fruits". The authors investigated a genome-wide analysis of Auxin Response Factor (ARF) genes in Rosa roxburghii and focused on RrARF5 and its regulatory role in ascorbic acid (vitamin C) biosynthesis. The study highlights the nutritional and commercial value of R. roxburghii. The manuscript is written well. However, some areas require revisions to improve scientific presentation of the manuscript.

Major Comments:

(1) Introduction should emphasize how this work advances current understanding.

(2) A phylogenetic tree was constructed using the neighbor-joining method. Maximum likelihood might yield more accurate evolutionary relationships is recommended.

(3) Were any of the RNA-seq findings validated via qRT-PCR beyond the transient expression system?

(4) The discussion on RrARF5’s mechanism of action could be extended. Compare the RrARF gene family features (e.g., intron-exon structures, motif distribution) with those from other Rosaceae members.

(5) Ensure that all supplementary figures (S1–S5) and tables referenced in the text are properly included and cited.

Minor Comments:

(6) Some minor language issues are present throughout the manuscript. Thoroughly revise the manuscript for grammar and typo errors.

(7) Line 18:"Cis-acting elements analyses revealed..." Sentence not clear. Improve it.

Example: "Analysis of cis-acting elements revealed..."

(8) Line 66: “…However, the role and mode of action of ARF in Vc biosynthesis and metabolism remain largely unknown.”

“ARF in Vc biosynthesis”  What is Vc?

(9) Line 78: “AsA accumulation is primary regulated…………………”

 “primary” should be “primarily”.

(10) Line 107: “...despite the ARF gene family is widely studied in other species…”

Incorrect grammar; should be “despite the fact that” or “although”.

(11) While the Introduction discusses auxin, ARF genes, and vitamin C biosynthesis in general, it does not clearly identify the specific gap in knowledge this study addresses. You can insert this sentence toward the end of your Introduction's background section to clearly define the gap that you want to address:

Example: “Despite ARF genes being well studied in several fruit crops, their genomic organization, expression profiles, and specific regulatory roles in vitamin C metabolism in Rosa roxburghii remain uncharacterized.”

(12) Line 121: “…using the 'Simple HMM search' in the TBtools software…”

"Simple HMM search" is not clear. What parameters were used? Was an E-value threshold set?

(13) Line 131: “...genes densities across the entire chromosomes were determined…”

 “Genes densities” should be “gene densities,”

(14) Line 399–401: “In the future, we will focus on the development of the plant genetic transformation technology of R. roxburghii and uncovering the biological function of ARF family members.”

(15)  “we will focus”  is not clear. It should be “Future research should focus on developing genetic transformation systems for R. roxburghii to experimentally validate the roles of ARF family members.”

(16) References: Some references use full journal names (e.g., this one), while others abbreviate (e.g., Mol. Plant). Use consistent journal name format.

Reviewer 3 Report

Comments and Suggestions for Authors

Rosa roxburghii Tratt is rich in vitamin C content and other nutritional compounds, has not yet been studied for its Auxin Responsive Factor (ARF) gene family, which plays a crucial role in plant growth and fruit development. In this study, the authors have  identiffed 14 ARF genes (designated as RrARFs) in R. roxburghii, which are distributed across seven chromosomes and grouped into four subfamilies. Cis-acting elements analyses revealed that these genes might be involved in various biological processes, including plant development, light responses, cell cycle regulation, phytohormone responses, and responses to abiotic and biotic stresses. Gene expression analysis demonstrated differential expression of RrARF genes across different tissues and stages of fruit development, with four members showing higher expression during the fruit ripening stages. The result from this study enhance our understanding of the ARF gene family's role in vitamin C accumulation in R. roxburghii and its broader impact on fruit quality regulation, thus, these results are original from R. roxburghii and results are significant.  In general, this manuscript was well-written and results are significant. I have the following comments:

Comments:

  1. For the key words, the first letter for each key word should be capital;
  2. In the introduction part, I don’t see any references cited from 2024-2025, the references need to be updated;
  3. Figure 2, the title with ARF family genes, ARF should be italic, please check it through the whole text;
  4. Figure 6, RrMDHAR1 gene. Gene should be italic;
  5. Figure 5B, why the overexpression is less than 3 times compare to the control?
  6. For the Figure 4 and Figure 5, there is no RT-PCR confirmations any these key results?

Round 2

Reviewer 3 Report

Comments and Suggestions for Authors

Since the authors have addressed all my questions and made the necessary corrections, thus, I have no further comments. I think the reversion is in a better form for publications.